# New Insights into Phase Separation Processes and Membraneless Condensates of EIN2

**DOI:** 10.3390/plants11162149

**Published:** 2022-08-18

**Authors:** Jian Lu, Chi-Kuang Wen, Georg Groth

**Affiliations:** 1National Key Laboratory of Plant Molecular Genetics, CAS Center for Excellence in Molecular Plant Sciences, Institute of Plant Physiology and Ecology, Chinese Academy of Sciences, Shanghai 200032, China; 2Institute of Biochemical Plant Physiology, Bioeconomy Science Center (BioSC), Cluster of Excellence on Plant Sciences (CEPLAS), Heinrich-Heine University Düsseldorf, D-40255 Düsseldorf, Germany

**Keywords:** liquid–liquid phase separation, membraneless condensates, ethylene signaling, EIN2, intrinsically disordered proteins

## Abstract

Recent technological advances allow us to resolve molecular processes in living cells with high spatial and temporal resolution. Based on these technological advances, membraneless intracellular condensates formed by reversible functional aggregation and phase separation have been identified as important regulatory modules in diverse biological processes. Here, we present bioinformatic and cellular studies highlighting the possibility of the involvement of the central activator of ethylene responses EIN2 in such cellular condensates and phase separation processes. Our work provides insight into the molecular type (identity) of the observed EIN2 condensates and on potential intrinsic elements and sequence motifs in EIN2-C that may regulate condensate formation and dynamics.

## 1. Introduction

The plant hormone ethylene is a key regulator of plant growth, development and stress adaption. Ethylene perception and response are mediated by a family of integral membrane receptors (ETRs) localized at the ER–Golgi network. These receptors, which form homo- and heteromers in their functional state at the ER membrane, act as negative regulators of the ethylene signaling pathway, following an inverse-agonist model in which ethylene binding switches off the downstream signal transmission [1,2,3]. Further studies have identified the ER-associated Raf-like kinase CTR1 and the ER integral membrane protein EIN2 as downstream elements of the receptors and integral parts of the ethylene signaling network [4,5]. Subsequent biochemical, molecular and cell biological work has provided a more detailed picture of the related molecular downstream signaling processes. These studies propose that exposure to ethylene switches the receptors off; in turn, CTR1 fails to inactivate EIN2. In consequence, EIN2 undergoes proteolytic cleavage, and the resulting EIN2 C-terminus (EIN2-C) is then transported to the nucleus where it activates ethylene signaling via the master transcription factor EIN3 and its paralogs by a mechanism that has to be further explored [6,7,8]. Subsequent studies disclosed that EIN2-activated ethylene signaling may further involve binding of ER-processed EIN2-C with the 3′-UTR of EIN3-binding F-box protein (EBF1/EBF2) transcripts and repression of their translation in P-bodies—membraneless cytoplasmic ribonucleoparticle (RNP) granules which assemble through liquid–liquid phase separation (LLPS) [9,10].

P-bodies, similar to other intracellular condensates, are dynamic structures maintained through multivalent interactions [11]. Recent technological advances allowing us to resolve molecular processes in living cells with high spatial and temporal resolution propose that such intracellular condensates are involved in diverse biological processes [11,12]. In total, we now know more than 10 different types of these heterogenous dynamic assemblies, usually composed of proteins and RNAs, that mediate functions from storage to transcriptional regulation [11,12]. Together, these diverse and versatile membraneless organelles show liquid-like behavior, adopt a spherical shape and undergo deformation and fusion events [13,14]. Furthermore, they all rapidly exchange components with the cellular milieu, and their properties are readily altered in response to environmental cues. The assembly of the diverse intracellular condensates that are known is probably encoded in their RNA and protein sequences [15,16], although our knowledge is still limited regarding which of the different components drive the phase separation process and what their specific roles are. Molecular flexibility seems to be a key parameter that facilitates intracellular condensate formation. Hence, intrinsically disordered proteins (IDPs) with their flexibility and polymer-like behavior qualify as essential elements in the formation of membraneless intracellular condensates [16,17]. In particular, prion-like domains (PLDs) enriched in polar amino acids such as asparagine, glutamine and serine are often found in IDPs together with blocks of arginine- and glycine-rich RNA-binding motifs [12,18,19]. Reconstitution experiments with corresponding sequences have provided strong evidence that prion-like IDPs drive the formation of many physiologically relevant intracellular condensates [16,17]. Based on these data, PLDs are emerging as important modules for gene regulation by reversible functional aggregation and phase transition.

In addition to the previously reported involvement of EIN2 in P-bodies (PBs) in response to ethylene, EIN2 is further found in cytoplasmic granules that localize to subcellular compartments distinct to the PB markers DCP1 and EIN5 [6,8]. This observation promoted us to investigate whether these EIN2 granules reflect a known type of phase separation condensate. Here, we report on the association of EIN2 with stress granules (SGs), agreeing with the sequence features of EIN2 potentially involving phase separation particles. Unexpectedly and interestingly, the translation of 10 gene sets involving a wide variety of LLPS particles was significantly impacted by the *EIN2* loss. Our findings highlight the possible involvement of EIN2 in a variety of the membraneless condensates as part of the underlying mechanism of ethylene signaling.

## 2. Results

### 2.1. Potential Intrinsic Elements and Sequence Motifs of EIN2 Critical for LLPS

The involvement of EIN2 in the EBF1/EBF2 translation repression in PBs turns EIN2 into a vital component of the PB condensate. To identify intrinsic elements in EIN2-C responsible for the observed phase separation of EIN2 bodies, and to pinpoint the related sequence motifs potentially involved in condensate formation, we have applied a range of sequence-based computational predictions. Results of these projections are illustrated in Figure 1A,B. In a nutshell, disorder prediction web-based servers DEPICTER [20] and PrDOS [21] suggest intrinsically disordered regions at aa 462–668, aa 711(717)–1047(1049), aa 1193(1196)–1213(1215) and aa 1238(1240)–1294. Web interface IUPred2A [22] providing context-dependent prediction of protein disorder as a function of redox state and protein binding is broadly in line with these predictions and proposes global folded domains at aa 667–735 and aa 969–1252. More specifically, the program PLAAC (prion-like amino acid composition [23]) pinpoints a single PLD in EIN2-C ranging from aa 775–792. Evidently, this sequence shows the typical over-representation of arginine or glutamine residues observed among PLDs and, thus, may be responsible to nucleate or drive phase separation of EIN2 bodies. Similarly, various approaches have been used in the past for predicting RNA-binding residues in proteins. Results from the DRNApred webserver [24], which are given in Figure 1C, indicate several putative RNA-binding residues in EIN2-C. RNABindRPlus [25]—a hybrid machine learning sequence homology-based approach—confirms that EIN2-C may have several putative RNA-binding sites. Previous experimental studies by the Guo lab [9] identified the NLS motif at aa 1262–1269 as critical for the translational repression function of EIN2 qualifying these residues—which are also predicted as putative binding sites by the DRNApred webserver— as excellent candidates for RNA-binding in EIN2-C. Intriguingly, studies by Gotor et al. (2020) found that a single PLD is sufficient to drive phase separation, whereas multiple RNA-binding domains (RBDs) are needed to confer liquid-like behavior and modulate the dynamics of the assemblies.

### 2.2. Association of EIN2 with Stress Granules and Translation Regulation

The identified EIN2 sequence elements indicative of LLPS are consistent with previous assumptions that the EIN2 granules observed in the cytoplasm may represent PBs for *EBF1*/*EBF2* translation repression. However, the reported cleavage complexity of the EIN2 protein raises concerns on the molecular type of the observed EIN2 granules [26,27]. These concerns are fueled by recent transient expression studies in *Arabidopsis* protoplasts revealing distinct localizations of EIN2 and the PB marker DCP1 [8]. Furthermore, in Arabidopsis transgenic plants, only expression of the full-length EIN2 caused formation of cytoplasmic granules, but not of the artificially defined EIN2-C that was processed into complex fragments [6,8].

Considering the possibility of false positives obtained from transient co-expression studies, partially due to excess proteins that are prone to non-specific aggregation or overflow, EIN2 sub-cellular localization was determined in stable transgenic plants. Arabidopsis wild-type (Col-0) lines expressing m*Cherry-G3BP7* and *DCP1-mChery*, respectively, were each genetically crossed with *EIN2-GFP* plants, and subcellular localizations of these proteins were determined by the laser scanning confocal microscopy (LSCM) technique from the resulting first filial generation (F1) plants. In line with the previous report of distinct localizations for EIN2 and the PB marker DCP1 in Arabidopsis protoplasts, the two proteins clearly did not co-localize in leaf cells of the F1 plants (Figure 2A). The Ras-GTPase-activating protein SH3-domain-binding protein G3BP7 is a scaffold protein for the assembly of stress granules (SGs), which can be induced by heat stress, and EIN2 and G3BP7 tightly colocalized upon a 42 °C induction in transgenic Arabidopsis plants co-expressing the two transgenes (Figure 2B). The EIN2 granules’ formation appeared independent of the SG marker upon the 42 °C induction in the *EIN2-GFP* transgenic plants (Figure 2C). These results agree with the sequence features of EIN2 potentially involving LLPS and do not exclude possible association of EIN2 with other LLPS condensates.

SGs may involve translation regulation and mRNA metabolism with PBs [28,29], and the association of EIN2 with gene sets prompted us to investigate translational changes involving LLPS on the loss of *EIN2*. The conventional gene ontology (GO) or KEGG analyses determine gene functions or biological pathways on the basis of a defined statistical significance level; however, biological functions or pathways are likely undermined when the involved genes are expressed in opposite directions or unchanged. These analytic challenges can be overcome by the gene set enrichment analyses (GESA) that evaluate the data at the gene set level [30,31]. Herein, gene sets involving LLPS were analyzed from the normalized ribosome foot-printing data of wild-type and the *ein2^W308^* early-termination mutant. Ten gene sets of significance (NOM *p* < 0.01, FDR q < 0.05) were determined, involving phase separation particles for wild-type but not for the *ein2^W308*^* seedlings (Appendix A). The prevalent down-regulation for translation of these gene sets in *ein2^W308*^* seedlings suggests pleiotropic impacts of the *EIN2* loss on those LLPS events, agreeing with roles of SGs in translation regulation.

## 3. Summary and Discussion

The present model for EIN2-activated ethylene signaling proposes that the EIN2 carboxyl portion (EIN2-C), processed at the ER, transports to the nucleus and activates ethylene signaling in the cytoplasm involving PBs via repressing the *EBF1/EBF2* mRNAs [9,10]. On the other hand, the complexity of EIN2 processing and subcellular localization adds uncertainty to the present molecular model of EIN2-activated ethylene signaling. The EIN2-C that can be processed into multiple fragments [8,27] is proposed to localize in the nucleus and target the *EBF1/EBF2* mRNAs in the cytoplasm [9,10]. In this regard, there may be a need for a signaling mechanism to export the nuclear EIN2-C to the cytoplasm in response to ethylene, and this nuclear–cytoplasmic transport is to be determined. Additionally, the magnitude of the contribution to ethylene signaling by the extent of *EBF1/EBF2* translation repression (approximately 50% reduction in the translation efficiency) has to be determined, as EBF1/EBF2 can also be regulated at the protein level, independent of the 3′-UTR targeting by EIN2 [32].

In contrast to the current model, our study revealed distinct subcellular localizations for EIN2 and PB markers, agreeing with a previous report involving transient expression in *Arabidopsis* protoplasts [8]. Unlike SGs, which are detectable under stress conditions, PBs are constitutively detectable by microscopy [29]. The cytoplasmic EIN2 granules observed that are distinct to PBs qualify as SGs—another type of membraneless condensate formed upon stress induction—and thus further support the intrinsic LLPS property of EIN2. It is conceivable that EIN2 can be dynamically recruited to SGs upon different type of inductions. Furthermore, it seems possible that EIN2 is recruited to other types of LLPS particles rather than SGs and PBs.

Our translatomics analyses further revealed involvement of EIN2 in translation of a wide variety of genes for LLPS. SGs are involved in translation regulation and mRNA homeostasis. Given that EIN2 can be recruited to SGs, which can be induced by stresses as well as plant hormones [28], it is conceivable that a basal level of EIN2 that is recruited to SGs may regulate translation of a subset of gene sets. Alternatively, the *EIN2* loss impacts translation of a subset of gene sets via a feedback loop. Interestingly, translation of the subset of genes for PBs and SGs also involves EIN2 (Supplementary Information), even though the two types of LLPS particles regulate translation by distinct mechanisms. The impact of *EIN2* loss on translation of the variety of gene sets may, in part, be attributed to the cascade of translation repression of the gene sets for PBs and SGs.

## 4. Materials and Methods

### 4.1. Laser Scanning Confocal Microscopy

The laser scanning confocal microscopy (LSCM) involved Leica (TSC SP8 STED3X) and Zeiss (LSM880). Arabidopsis transgenic lines expressing m*Cherry-G3BP7* and *DCP1-mChery* were genetically crossed with *EIN2-GFP* plants, and subcellular localizations of these proteins were determined by the LSCM from leaf or seedling hypocotyl cells of the resulting plants.

### 4.2. Translatomes and Gene Set Enrichment Analysis (GSEA)

Seedlings of wild-type Arabidopsis (Col-0) and the *ein2^W308^* mutants were grown under light (16 hr light and 8 hr dark) and harvested 7 days after germination for translatomics analyses. The translatomics, involving RNA- and Ribo-seq, and data analyses were a customer service by Genedenovo Biotechnology (Guangzhou, China). Normalized ribo-deq data (FKPM) were subject to GSEA as instructed by GSEA (gsea-msigdb.org, accessed on 21 December 2021) [31], and Arabidopsis gene sets involving LLPS were downloaded from PLANTGSED [32].

## Figures and Tables

**Figure 1 plants-11-02149-f001:**
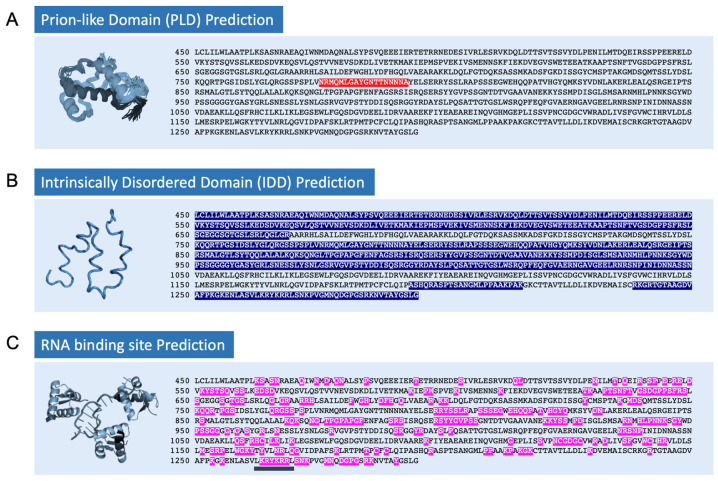
**Sequence-based computational studies on the involvement of EIN2 in liquid–liquid phase separation (LLPS).** Sequence-based analysis of potential residues and domains in EIN2-C that may be involved in phase separation. (**A**) Prion-like domains predicted by *PLAAC* (highlighted in red), (**B**) intrinsically disordered domains identified by *PrDOS*, *DEPICTER* and *IUPred2A* web-based server (highlighted in blue), (**C**) putative RNA-binding residues computed by the *DRNApred* webserver (highlighted in pink). Experimentally validated RNA-binding site in EIN2-C at 1262–1269 is underlined in black.

**Figure 2 plants-11-02149-f002:**
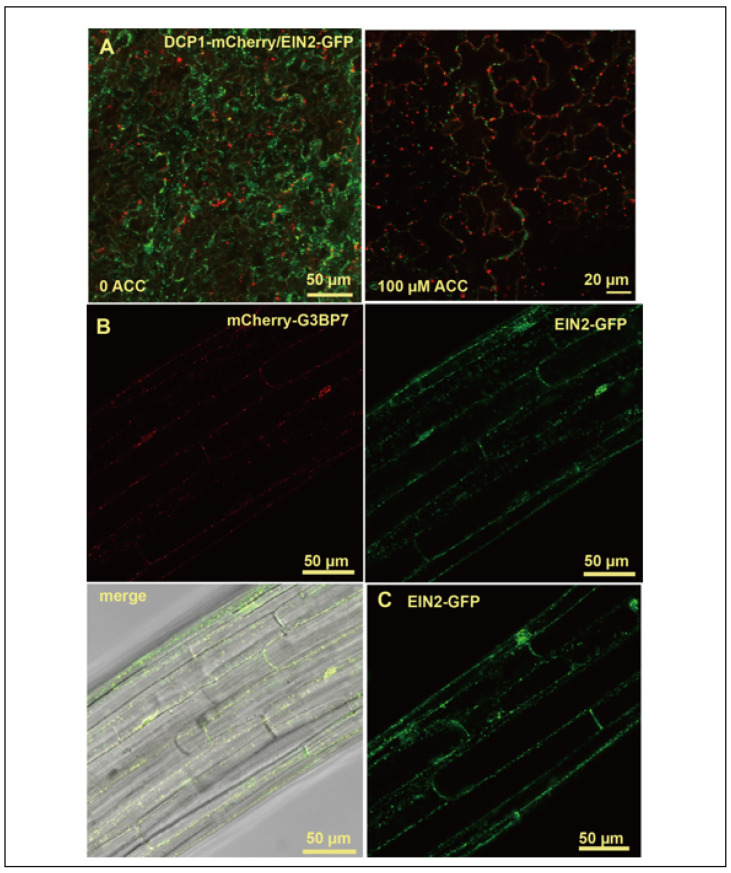
**Molecular experimental studies on the involvement of EIN2 in LLPS.** Subcellular localizations of EIN2-GFP with LLPS markers. (**A**) Fluorescence of EIN2-GFP (green) and PB marker DCP1-mCherry (red), without (0 ACC) and with ACC (100 μM) treatment. (**B**) Fluorescence of the SG marker mCherry-G3BP1 (red) and EIN2-GFP (green) upon a 42 °C stress induction. (**C**) The EIN2-GFP granule formation was independent of the SG marker upon the heat induction. The images were acquired from leaf (**A**) and seedling hypocotyl (**B**,**C**) cells of *Arabidopsis* transgenic lines expressing transgenes encoding the indicated proteins.

## Data Availability

The ribo-seq data are deposited at the BioSample database (https://www.ncbi.nlm.nih.gov/bioproject/861101), BioProject ID PRJNA861101. BioSample accessions: SAMN29883430, SAMN29883431, SAMN29883432, SAMN29883433, SAMN29883434, SAMN29883435.

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
