# Peer review of "New Insights into Phase Separation Processes and Membraneless Condensates of EIN2"

_plants, 2022, doi:10.3390/plants11162149_

Round 1

Reviewer 1 Report

Nice set of data suggesting that EIN2 participates to liquid-liquid phase separation  via translation regulations and that part of it may be localized in stress granules.

line 82: please, replace « constitutient » by « « component »

line 104: please, spell out RBDs

line 107: spell out LLPS, so the figure legend can stand by itself

 lines 170 to 184: the use of “GSs” instead of “gene sets” makes the discussion hard to follow, as it is confusing in parallel to the use of "SGs". I suggest to the authors to avoid the "GS" acronym and to sometimes replace “gene sets” by “genes”, when it is possible.

Author Response

Nice set of data suggesting that EIN2 participates to liquid-liquid phase separation via translation regulations and that part of it may be localized in stress granules.

We thank the Reviewer for this positive assessment of our manuscript.

line 82: please, replace « constitutient » by « « component »

line 104: please, spell out RBDs

line 107: spell out LLPS, so the figure legend can stand by itself

Changes have been made accordingly.

lines 170 to 184: the use of “GSs” instead of “gene sets” makes the discussion hard to follow, as it is confusing in parallel to the use of "SGs". I suggest to the authors to avoid the "GS" acronym and to sometimes replace “gene sets” by “genes”, when it is possible.

We have modified the text in accordance with the Reviewer’s suggestions, removed the akronym GS and replaced "gene sets" by "genes" where possible.

Reviewer 2 Report

This work performed bioinformatics analysis on a key regulator of ethylene signaling EIN2 for its RNA binding potential and the putative domain for LLPS. The work experimentally validated the role of EIN2 for its role in forming LLPS in association with stress granule rather than previously reported processing body. Subsets of SG related mRNAs was identified as translationally regulated by EIN2 by ribosome sequencing experiment in the ein2 mutant compared with Col-0 wildtype. Therefore the study concluded that there is a role of translational control of EIN2 through LLPS by SG.

Major points:

1.       The bioinformatics analysis of EIN2 for its potential for LLPS and RNA binding is interesting. However the experiment performed do not validated either the RNA binding potential by for example identify the RNA binding targets or the protein sequence that control the LLPS formation as the prediction for example by mutation analysis. Therefore, no direct link could be made between the bioinformatics analysis and experiment validation. Mutation on the identified domain for PPLS detection would be interesting functional validation.

2.       The ribosome sequencing experiment is performed between ein2W308* and Col-0 seedlings. However no sequencing quality information is provided such as the periodicity and reads distribution were provided, which is essential for the success of ribosome sequencing library preparation. Therefore it is not possible to evaluate the quality of the ribosome experiment.

3.       Line 153, when talking about the down-regulation of the GSs in ein2W308* , it’s not known what the this refers to. Is it for down-regulation of translatome (the ribosome associated mRNA), or translational efficiency (ribosome RNA/total RNA)? Substantial details are needed to for the data analysis on the ribosome sequencing data since a changed translatome do not mean a role of translational control. A changed translational efficiency does.

Minor:

1.       Figure 2 figure quality should be improved. There are duplicates for the scale bar for each graph and the fluorescence signal should be adjusted for better visualization (maybe increase the dose?) .

2.       Ribose data should be submitted to database for public access.

Author Response

This work performed bioinformatics analysis on a key regulator of ethylene signaling EIN2 for its RNA binding potential and the putative domain for LLPS. The work experimentally validated the role of EIN2 for its role in forming LLPS in association with stress granule rather than previously reported processing body. Subsets of SG related mRNAs was identified as translationally regulated by EIN2 by ribosome sequencing experiment in the ein2 mutant compared with Col-0 wildtype. Therefore the study concluded that there is a role of translational control of EIN2 through LLPS by SG.

Major points:

  1. The bioinformatics analysis of EIN2 for its potential for LLPS and RNA binding is interesting. However the experiment performed do not validated either the RNA binding potential by for example identify the RNA binding targets or the protein sequence that control the LLPS formation as the prediction for example by mutation analysis. Therefore, no direct link could be made between the bioinformatics analysis and experiment validation. Mutation on the identified domain for PPLS detection would be interesting functional validation.

We thank the Reviewer for the encouraging comment on our work and we fully agree that the suggested mutation analysis would add further value to our work. However, this type of experiments requires recombinant production of wild type EIN2-C and a set of mutants in the bacterial expression system used. In previous studies we found that purity and yield of EIN2-C are substantially affected by mutations. As a matter of fact, EIN2-C is difficult to purify at sufficient quantities and large concentration for an in vitro phase separation assay. The EIN2-C purified from our bacterial expression system is quite aggregation-prone, difficult to purify and to handle. Besides, comprehensive systematic studies will be necessary to identify suitable conditions for in vitro phase separation of recombinant purified EIN2-C (variations in pH, salt concentration, temperature, solubilizing agents etc.). In addition to the aforementioned technical challenges, EIN2-C or EIN2 have been reported to be cleaved into complex fragments (1-3). Although Qiao et al. (2) proposed the complex cleavages resulting from degradation during the 100°C SDS solubilization, this argument was not conclusive and can be argued from a different perspective that the 65°C SDS solubilization may lead to degradation of those cleaved EIN2-C and EIN2. Besides, our unpublished data did not find difference in EIN2-C/EIN2 cleavages at the two different SDS-solubilization conditions. Determining the fragments of EIN2-C that involve the granule formation will be the first step to identify domains involving the LLPS. The tight association of EIN2 with the SG marker upon heat induction may support the property of EIN2 being likely involving LLPS. On the other hand, follow-up extensive, in-depth experiments determining domains involving the LLPS process will be needed while being beyond the scope of this report.

  1. The ribosome sequencing experiment is performed between ein2W308* and Col-0 seedlings. However no sequencing quality information is provided such as the periodicity and reads distribution were provided, which is essential for the success of ribosome sequencing library preparation. Therefore it is not possible to evaluate the quality of the ribosome experiment.

The Ribo-seq quality data information is attached (Files: Ribo-seq quality data.pdf). These quality features may ensure the subsequent translatomics analyses.

  1. Line 153, when talking about the down-regulation of the GSs in ein2W308* , it’s not known what the this refers to. Is it for down-regulation of translatome (the ribosome associated mRNA), or translational efficiency (ribosome RNA/total RNA)? Substantial details are needed to for the data analysis on the ribosome sequencing data since a changed translatome do not mean a role of translational control. A changed translational efficiency does.

      This is an important comment. In this study, the ribosome-associated mRNA, instead of the TE (translation efficiency), were analyzed. The translation instead of the TE for gene sets involving LLPS was analyzed in this study because it is the level of translating mRNAs but not TE that may reflect the level of de novo biosynthesized proteins (without considering other factors that may affect the steady-state protein level). Besides, the TE does not necessarily determine a direct translational control by EIN2; there may be higher order regulations, and the complexity cannot be simply determined by the TE or translatomics. We did not intend to address issues about direct translational control by EIN2; the present data simply suggested the translation involve EIN2. In this manuscript, we did not make inference for a direct control of EIN2 on translation of those gene sets, and it was “translation” instead of translation efficiency described in the manuscript. We thus discussed several possibilities for the involvement of EIN2 in translation of those gene sets.

Minor:

  1. Figure 2 figure quality should be improved. There are duplicates for the scale bar for each graph and the fluorescence signal should be adjusted for better visualization (maybe increase the dose?) .

The images were replaced with ones with a better resolution.

  1. Ribose data should be submitted to database for public access.

         Thanks for this comment. This work was not intended to compare genome-wide translatomes for wild type and the ein2 mutant, and it was why the complete set of data is not included. Besides, according to the publication policy, we do not find this to be mandatory.  If reviewer 2 insist, we will be pleased to upload the complete sets of data (gene sets in excel format) for the GSEA as supplementary information.

Reference:

  1. Zhang J, Chen Y, Lu J, Zhang Y, & Wen C-K (2020) Uncertainty of EIN2Ser645/Ser924 Inactivation by CTR1-Mediated Phosphorylation Reveals the Complexity of Ethylene Signaling. Plant Communications 1(3).
  2. Qiao H, et al. (2013) Response to Perspective. Plant Signaling & Behavior 8(8):e25037.
  3. Wen X, et al. (2012) Activation of ethylene signaling is mediated by nuclear translocation of the cleaved EIN2 carboxyl terminus. Cell research 22(11):1613-1616.

Round 2

Reviewer 2 Report

Thanks for providing the quality data for ribo-seq.

Here are a few concerns related to this dataset.

1.  A paired-end ribo-seq library is applied based on that the adapters were addded on both sides, I am wondering why since the ribo-seq library is small (about 28bp)and single-end linker ligation is normally applied with further circularization step for library amplification, especially when you refer to the Ingolia NT, 2012.

2. The periodicity plot from meta periodicity doesn't look like the data is periodic. The reason could be: You used the reads range between 20-40bp for the ribo-seq analysis, which is very broad read length range and it will bring lots of noise and background from the ribo-seq data. Then the periodicity is diluted due to the noisy reads that are not actual ribo-reads. I suggest using either rends_heat or frame_psite_length from riboWaltz to visualize your data and identify which length follow the periodic pattern and either use that length only or pooled the 1-2 periodic length reads for the follow-up analysis. This will largely remove the non-periodic noise reads.

3. Figure legend is missing in the last subfigure on data quality. Figure should be self-explained.

Author Response

  1. A paired-end ribo-seq library is applied based on that the adapters were addded on both sides, I am wondering why since the ribo-seq library is small (about 28bp)and single-end linker ligation is normally applied with further circularization step for library amplification, especially when you refer to the Ingolia NT, 2012.

>Thanks for this comment. Yes, we agree with the comment that a single-end sequencing will be sufficient to reading through, with similar results obtained from a paired-ended sequencing. The pair-ended sequencing was applied simply for the purpose of elevating ligation efficacy when both ends were sequenced. Besides, it will be more economic (lower cost) for the pair-ended sequencing.

  1. The periodicity plot from meta periodicity doesn't look like the data is periodic. The reason could be: You used the reads range between 20-40bp for the ribo-seq analysis, which is very broad read length range and it will bring lots of noise and background from the ribo-seq data. Then the periodicity is diluted due to the noisy reads that are not actual ribo-reads. I suggest using either rends_heat or frame_psite_length from riboWaltz to visualize your data and identify which length follow the periodic pattern and either use that length only or pooled the 1-2 periodic length reads for the follow-up analysis. This will largely remove the non-periodic noise reads.

>Thanks for this comment that helps improve the data presentation. Here riboWaltz was involved to generate the periodicity.

  1. Figure legend is missing in the last subfigure on data quality. Figure should be self-explained.

>We revised the figure and figure legends for supplementary figure S2.

Round 3

Reviewer 2 Report

My previous comments are only partly addressed. 

It is still not clear in your Figure S2G-F what these periodicity plot based on, from the specific RF length range or all the RF? From your length distribution in Figure S2B, your length range is from 20-40 bp. There are abundant RF of these that are simply RNA digestion fragments that are not associated with ribosomes. If you take them into your follow-up calculation, it will bring lots of artifacts. The normal way of processing ribosome data is to select a specific RF length (normally 28bp or range that show periodicity) rather than a pool of the RF from all lengths (see Basin et al, 2017, PNAS). You can visualize the periodicity of a specific length with ribowalts as I mentioned before and then make the decision on which length to include. You didn't label what the color of each bar represents in the periodicity plot.
